# Mertk Interacts with Tim-4 to Enhance Tim-4-Mediated Efferocytosis

**DOI:** 10.3390/cells9071625

**Published:** 2020-07-06

**Authors:** Byeongjin Moon, Juyeon Lee, Sang-Ah Lee, Chanhyuk Min, Hyunji Moon, Deokhwan Kim, Susumin Yang, Heera Moon, Jaeseon Jeon, Young-Eun Joo, Daeho Park

**Affiliations:** 1School of Life Sciences, Gwangju Institute of Science and Technology, Gwangju 61005, Korea; byeongjinmoon@gist.ac.kr (B.M.); iris260@gist.ac.kr (J.L.); sanga03@gist.ac.kr (S.-A.L.); alscksgur@gist.ac.kr (C.M.); hjmoon311@gist.ac.kr (H.M.); po7322@gist.ac.kr (D.K.); susuminy@gist.ac.kr (S.Y.); gmlfk301@gist.ac.kr (H.M.); jeonjaeseon17@gist.ac.kr (J.J.); 2Center for Cell Mechanobiology, Gwangju Institute of Science and Technology, Gwangju 61005, Korea; 3Department of Internal Medicine, Chonnam National Univerity, Gwangju 61469, Korea; yejoo@chonnam.ac.kr

**Keywords:** efferocytosis, Tim-4, Mertk, engulfment, apoptosis, phosphatidylserine, receptor

## Abstract

Apoptotic cells expressing phosphatidylserine (PS) on their cell surface are directly or indirectly recognized by phagocytes through PS-binding proteins. The PS-binding protein Tim-4 secures apoptotic cells to phagocytes to facilitate the engulfment of apoptotic cells. However, the molecular mechanism by which Tim-4 transduces signals to phagocytes during Tim-4-mediated efferocytosis is incompletely understood. Here, we report that Tim-4 collaborates with Mertk during efferocytosis through a biochemical interaction with Mertk. Proximal localization between the two proteins in phagocytes was observed by immunofluorescence and proximal ligation assays. Physical association between Tim-4 and Mertk, which was mediated by an interaction between the IgV domain of Tim-4 and the fibronectin type-III domain of Mertk, was also detected with immunoprecipitation. Furthermore, the effect of Mertk on Tim-4-mediated efferocytosis was abolished by GST-Mertk^FnIII^, a soluble form of the fibronectin type-III domain of Mertk that disrupts the interaction between Tim-4 and Mertk. Taken together, the results from our study suggest that a physical interaction between Tim-4 and Mertk is necessary for Mertk to enhance efferocytosis mediated by Tim-4.

## 1. Introduction

Apoptotic cells generated during development and in homeostasis are efficiently removed by a process called efferocytosis [1]. To facilitate this process, phagocytes have developed a unique apparatus to distinguish apoptotic cells from live cells and to specifically phagocytose apoptotic cells [2,3]. Phosphatidylserine (PS) exposed on the outer leaflet of the plasma membrane is a key molecule utilized by the apparatus. PS exposed on apoptotic cells is recognized by phagocytes through PS-binding proteins, either directly or indirectly [4,5,6]. PS-binding proteins can be cytoplasmic proteins, transmembrane proteins, or secreted proteins. Transmembrane and secreted PS-binding proteins are the types predominately involved in recognition of apoptotic cells by phagocytes due to their subcellular locations and ability to bind to PS on apoptotic cells [7].

Since PS expression on apoptotic cells is indispensable for efferocytosis [8], studies to identify PS receptors have been carried out over the past couple of decades, and have identified several engulfment receptors for apoptotic cells. Some of these receptors are referred to as PS receptors, including Bai1, Sabilin-2, and Tim-4, which directly recognize PS on apoptotic cells [9,10,11,12,13]. Other engulfment receptors, such as TAM receptors (for example Mertk) and integrins, indirectly sense PS on apoptotic cells through recognition of secreted PS-binding proteins called bridging molecules [14,15,16,17]. PS receptors have differing abilities to transduce signals into phagocytes. Some PS receptors can directly transduce signals into phagocytes after binding to apoptotic cells, while other receptors secure apoptotic cells to phagocytes without directly initiating signal transduction; these latter receptors are called tethering receptors [18,19,20]. Tethering receptors are thought to facilitate the recognition and ingestion of apoptotic cells by other engulfment receptors capable of mediating direct signaling [19,21,22,23,24].

Among PS receptors, Tim-4 has received much attention due to its relevance to allergic diseases, its high expression in macrophages, and the availability of the crystal structure of the Tim-4–PS complex [11,25,26,27,28]. Initially, Tim-4 was identified as a ligand for Tim-1 that regulates T-cell proliferation [27]. Later, it was reported that Tim-4 binds to PS expressed on apoptotic cells and thus functions as a PS receptor [11]. Tim-4, however, does not mediate direct signaling in efferocytosis because its cytoplasmic tail, which is about 40 amino acids long, is too short to contain a signaling motif [29]. Moreover, efferocytosis mediated by Tim-4 mutants lacking the cytoplasmic tail or transmembrane domain is unimpaired and comparable with that mediated by the full-length protein [18,19,23]. Thus, current understanding is that Tim-4 secures apoptotic cells to phagocytes by directly binding to PS on apoptotic cells, and that other engulfment receptors such as α_v_β_3_ and Mertk that are capable of direct signaling thereafter efficiently recognize apoptotic cells and mediate their ingestion.

We recently reported how Tim-4 collaborates with integrins that can directly transduce signals. The IgV domain of Tim-4 interacts with the fibronectin type-III (FnIII) domain of Fn1, and thus Fn1 forms a complex with Tim-4 and integrins. The formation of this complex promotes synergistic cooperation of these two types of engulfment receptors, leading to efficient binding to and ingestion of apoptotic cells [19]. However, disrupting the complex or inhibiting integrins partially inhibits Tim-4-mediated efferocytosis. Interestingly, Mertk, which also functionally collaborates with Tim-4 and promotes Tim-4-mediated efferocytosis, possesses two FnIII domains in its extracellular region [21,30,31]. These properties of Mertk led us to validate the biochemical interaction between Tim-4 and Mertk and, if any, the effect of the interaction on Tim-4-mediated efferocytosis. Our results showed that Tim-4 and Mertk were located in close proximity and biochemically interacted with each other. The interaction between Tim-4 and Mertk was mediated by association of the IgV domain of Tim-4 with the FnIII domain of Mertk. In addition, the enhancing effect of Mertk on Tim-4-mediated efferocytosis was nullified by GST-Mertk^FnIII^, a soluble form of the FnIII domains of Mertk that disrupts the interaction between Tim-4 and Mertk. GST-Mertk^FnIII^ impaired efferocytosis mediated by peritoneal macrophages derived from wild-type (*WT*) mice but did not impair efferocytosis mediated by peritoneal macrophages derived from *Tim-4^-/-^* or *Mertk^-/-^* mice. Taken together, the results of our study suggest that a physical association between Tim-4 and Mertk is necessary for Mertk to enhance Tim-4-mediated efferocytosis.

## 2. Materials and Methods

### 2.1. Cell Culture and Transfection

293T cells were maintained in DMEM (Dulbecco’s Modified Eagle’s Medium) containing 10% FBS (Fetal Bovine Serum) and 1% PSQ (Penicillin-Streptomycin-Glutamine). LR73 cells were maintained in Alpha-MEM (Alpha’s Modified Eagle’s Medium) containing 10% FBS and 1% PSQ. 293T cells were transfected by using calcium phosphate transfection system (Promega, Madison, WI, USA) and LR73 cells were transfected by using Lipofectamin 2000 (Invitrogen, Waltham, MA, USA) according to the manufacturer’s protocol.

### 2.2. Plasmids and Antibodies

All plasmids made in this study were generated by a cloning strategy based on PCR and sequenced to confirm their accuracy of sequence. Mouse Mertk cDNA were purchased from Open Biosystem (Cat #: OMM5896-202524955). HA-Tim-4, Tim-4^ECR^-FLAG, Tim-4^IgV^, and Tim-4^mucin^ were previously used [18]. Mertk^ECR^, Mertk^Ig^, and Mertk^FnIII^ constructs contain residues 19-497, 19-277, and 278-497, respectively. The antibodies used in this study were anti-FLAG (F1804, Sigma Aldrich, St. Louis, MO, USA), anti-HA (SC-7392, Santa Cruz biotechnology, Dallas, TX, USA), anti-HA (#3724, Cell signaling technology, Danvers, MA, USA), anti-GFP (ab290, Abcam, Cambridge, MA, USA), anti-GST (SC-138, Santa Cruz biotechnology, Dallas, TX, USA), anti-mouse Mertk (AF591, R&D Systems, Minneapolis, MN, USA), anti-Tim-4 (SC-79143, Santa Cruz biotechnology, Dallas, TX, USA), anti-Tim-4 (ab176486, Abcam, Cambridge, MA, USA), anti-Actin (SC-47778, Santa Cruz biotechnology, Dallas, TX, USA), and normal goat IgG control (AB-108-C, R&D Systems, Minneapolis, MN, USA). Fluorochrome-conjugated donkey anti-goat secondary antibody (Alexa Fluor 488, A-11055) and goat anti-rabbit secondary antibody (Alexa Fluor 594, A-11037, Alexa Fluor 405, A-31556) were purchased from Thermo Fisher Scientific (Carlsbad, CA, USA).

### 2.3. Mice

*Tim-4^-/-^* mice (RBRC04895) were obtained from Riken BioResource Center (Japan), *Mertk^-/-^* mice (011122) were purchased from Jackson Laboratories (Bar Harbor, USA). All experiments using mice were approved by the animal care and ethics committees (LARC) of the Gwangju institute of science and technology (GIST) in accordance with the national institutes of health guide for the care and use of laboratory animals.

### 2.4. Immunoprecipitation and Immunoblotting

293T cells were transfected with the indicated plasmids. Then, 2 d after transfection, the cells were lysed, incubated with appropriate antibodies with protein A/G-conjugated (10001D, 10003D, Thermo Fisher Scientific, Carlsbad, CA, USA), FLAG-conjugated (A2220, Sigma Aldrich, St. Louis, MO, USA), or glutathione agarose beads (17-0756-01, GE healthcare, Chicago, IL, USA) at 4 °C for 2 h. Bound proteins were separated on SDS-PAGE, transferred onto nitrocellulose membrane, and detected with pertinent antibodies.

### 2.5. Immunostaining

LR73 cells were cultured on 18 mm Φ coverslips coated with poly-D-Lysine, on a 12-well non-culture plate for 36 h and transfected. The day after transfection, the cells were washed with PBS, fixed with 4% paraformaldehyde in PBS for 15 min. After fixation, cells were blocked with 10% BSA in PBS for 30 min and incubated with primary antibody, anti-HA, anti-Mertk in 3% BSA in PBS at 4 °C overnight. Then cells were washed with PBS for 5 min twice and stained with Alexa fluor 405, 488, and 594 conjugated secondary antibodies for 1 h. Images were obtained using confocal microscope (FV1000 SPD, Olympus, Tokyo, Japan). The co-localization index was calculated as follows. The formula for the co-localization index is the number of pixels with a co-localization color divided by the number of total pixels. The range of two colors from green (Mertk) to red (Tim-4) was set, and the pixels in the middle range of the color (yellowish) was counted using ImageJ 1.4.

### 2.6. Efferocytosis Assay

Efferocytosis assays were performed as previously described [32]. Briefly, LR73 cells were transfected with the indicated plasmids. Then, 1 d after transfection, the cells were preincubated in serum-free alpha-MEM for 2 h and then incubated with 5-(and-6)-Carboxytetramethylrhodamine (TAMRA)-labelled apoptotic thymocytes suspended in serum-free MEM in the presence or absence of purified proteins for 2 h. The cells then were extensively washed with ice-cold PBS, trypsinized, and analyzed using flow cytometry (BD FACS Canto II). Data from flow cytometry were analyzed by FLOWJO software (FlowJo LLC, Ashland, OR, USA). For assays for efferocytosis by peritoneal macrophages, peritoneal macrophages derived from indicated mice were incubated with TAMRA-labelled apoptotic thymocytes for 15 min, then washed with ice-cold PBS, trypsinized, and analyzed using flow cytometry.

### 2.7. Proximity Ligation Assay

Proximity ligation assays were performed according to the manufacturer’s protocol (DUO92101, DUO92003, Sigma Aldrich, St. Louis, MO, USA). Briefly, LR73 cells were cultured on 18 mm Φ coverslips coated with poly-D-Lysine and transfected with indicated plasmids. Then, 1 d after transfection, the cells were fixed with 4% paraformaldehyde in PBS, permeabilized with 0.1% PBST, blocked with Duolink^®^ blocking solution for 1 h, and incubated with anti-FLAG (F1804, Sigma Aldrich, St. Louis, MO, USA) and anti-Tim-4(SC-79143, Santa-Cruz biotechnology, Dallas, TX, USA) antibodies at 4 °C overnight. Then, the cells were incubated with anti-mouse and anti-goat PLA probes, and then ligation and amplification were sequentially allowed to occur in ligation solution and amplification buffer. Then the cells were fixed, and images were acquired using confocal microscopy.

### 2.8. Statistical Analysis

All data are shown as the mean value ± standard deviation. Each experiment was performed independently at least two times. Statistical significance was analyzed by two-tailed *t*-test in GraphPad Prism 7 software and significance was accepted when *p* values were < 0.05

## 3. Results

### 3.1. Tim-4 is Co-Localized with Mertk

Previously, we showed that during Tim-4-mediated efferocytosis, Fn1 functions as a scaffold by bringing together the IgV domain of Tim-4 and the FnIII domain of Fn1. However, abrogating the interaction of Tim-4 with Fn1 or blocking integrin signaling only partially inhibited Tim-4-mediated efferocytosis [19], implying redundancy of signaling in Tim-4-mediated efferocytosis. Interestingly, Mertk possesses two FnIII domains in its extracellular region (Figure 1A), and it is reported that Mertk functionally collaborates with Tim-4 to facilitate engulfment of apoptotic cells [21,30]. We also confirmed the effect of Mertk on Tim-4-mediated efferocytosis under serum-free conditions, which did not contain bridging molecules such as Gas6 required for Mertk-mediated efferocytosis and thus excluded the effect of Mertk itself on efferocytosis: efferocytosis mediated by cells overexpressing both Tim-4 and Mertk was superior to that by cells overexpressing Tim-4 alone (Figure 1B), and this superior efferocytosis was not due to an increase in the number of apoptotic cells binding to phagocytes (Figure 1C). These findings led us to test the possibility that Mertk could relay signals from Tim-4-tethered apoptotic cells into phagocytes through a physical interaction with Mertk. To address this possibility, the close physical distance between Tim-4 and Mertk was validated using two approaches, immunofluorescence and proximity ligation assays. First, FLAG-tagged Mertk was expressed in LR73 cells stably expressing hemagglutinin (HA)-tagged Tim-4, and their co-localization was evaluated using confocal microscopy. Both Tim-4 and Mertk were localized in the plasma membrane and fluorescence emitted from them was substantially superimposed, distinctly showing co-localization of Tim-4 and Mertk (Figure 1D). Next, we tested whether they are located in close enough proximity to physically interact with each other, using a proximal ligation assay. Fluorescence signals were only observed in LR73 cells co-expressing Tim-4 and Mertk, whereas signals were not detected in cells expressing either Tim-4 or Mertk alone (Figure 1E). These data indicate that a biochemical interaction between Tim-4 and Mertk is feasible.

### 3.2. Tim-4 Interacts with Mertk

Next, to directly test whether Tim-4 biochemically interacts with Mertk, HA-Tim-4 and Mertk-FLAG were overexpressed in 293T cells, and immunoprecipitation assays were performed. HA-Tim-4 was robustly co-precipitated with Mertk-FLAG by an anti-FLAG antibody (Figure 2A). This co-precipitation was inversely observed as well; Mertk-FLAG was co-precipitated with HA-Tim-4 by an anti-Tim-4^tail^ antibody (Figure 2B). Furthermore, to test whether the interaction could occur at endogenous protein levels, peritoneal macrophages, known to endogenously express both Tim-4 and Mertk, were lysed and Tim-4 was immunoprecipitated with an anti-Tim-4^tail^ antibody. Similarly, Mertk co-precipitated with Tim-4 at endogenous protein levels (Figure 2C), suggesting that Tim-4 physically interacts with Mertk. It is known that Mertk is phosphorylated during efferocytosis [30]. We thus tested whether Mertk phosphorylation could affect the interaction between Tim-4 and Mertk. Treatment of cells overexpressing Mertk with LDC1267 drastically reduced Mertk phosphorylation, but it unaltered the interaction between the proteins (Appendix A).

Unexpectedly, however, antibodies recognizing the extracellular region of Tim-4 or Mertk failed to co-precipitate Mertk or Tim-4, respectively, although they precipitated their own antigens (Figure 2D,E). These findings might result from steric hindrance caused by the antibodies, and also implies that the extracellular regions of Tim-4 and Mertk might mediate the interaction between Tim-4 and Mertk.

### 3.3. The IgV Domain of Tim-4 Binds to the Fibronectin Type III Domain of Mertk

To test whether the extracellular regions of Tim-4 and Mertk indeed mediate the interaction between Tim-4 and Mertk, Tim-4 lacking both the cytoplasmic tail and the transmembrane domain (HA-Tim-4-GPI) and Mertk-FLAG were expressed in 293T cells, and the interaction between the proteins was evaluated. In these experiments, HA-Tim4-GPI levels were always lower than HA-Tim-4 when Mertk was co-expressed; the reasons for this are unknown. Nevertheless, HA-Tim-4-GPI was robustly co-precipitated with Mertk by an anti-FLAG antibody (Figure 3A), suggesting that Tim-4 and Mertk interact through their extracellular regions. We thus tested whether an interaction occurred between the extracellular regions of Tim-4 and Mertk. The extracellular region of Tim-4 co-precipitated with the extracellular region of Mertk (Figure 3B). Since we previously found that the IgV domain of Tim-4 interacts with the FnIII domain of Fn1, we further tested whether the IgV domain of Tim-4 and the FnIII domain of Mertk mediate the interaction between Tim-4 and Mertk. As expected, the FnIII domain of Mertk co-precipitated with the extracellular region of Tim-4, whereas the Ig domain of Mertk did not co-precipitate (Figure 3C). Inversely, the IgV domain of Tim-4 co-precipitated with the extracellular region of Mertk, while the mucin domain of Tim-4 did not co-precipitate (Figure 3D and Appendix A). In agreement with a previous report [19], the expression level of the IgV domain of Tim-4 was exceptionally low. Thus, the IgV domain was undetectable in total cell lysates. However, it was detected in immunoprecipitants when the protein was precipitated and concentrated by an anti-FLAG antibody (Figure 3E), suggesting that the extracellular region of Mertk strongly interacts with the IgV domain of Tim-4. The interaction between the IgV domain of Tim-4 and the FnIII domain of Mertk was further confirmed (Figure 3F). Taken together, these data indicate that the IgV domain of Tim-4 and the FnIII domain of Mertk mediates the interaction between Tim-4 and Mertk.

### 3.4. Disrupting the Interaction between Tim-4 and Mertk Abolishes the Effect of Mertk on Tim-4-Mediated Efferocytosis

Next, we validated the importance of the physical interaction between Tim-4 and Mertk for their synergistic effects on efferocytosis. Due to the necessity of the FnIII domains of Mertk for the interaction between Mertk and Tim-4, we generated a soluble form of the FnIII domain of Mertk (GST-Mertk^FnIII^) from 293T cells and validated whether the soluble protein could disrupt the interaction of Tim-4 with Mertk. Notably, the interaction between Tim-4 and Mertk was attenuated in the presence of GST-Mertk^FnIII^ compared with the same interaction in the presence of GST (Figure 4A and Appendix A). Furthermore, the fluorescence signal for co-localization of Tim-4 with Mertk was weaker in cells treated with GST-Mertk^FnIII^ than in cells treated with GST (Figure 4B), and co-localization was significantly reduced by GST-Mertk^FnIII^ (Figure 4C and Appendix A). Additionally, we tested whether the soluble protein affects the interaction between Tim-4 and Mertk using PLA assay. The PLA fluorescence from cells expression Tim-4 and Mertk in the presence of GST-Mertk^FnIII^ was noticeably lower than that from control cells (Appendix A) suggesting that GST-Mertk^FnIII^ impedes the interaction between Tim-4 and Mertk. We then tested the effects of GST-Mertk^FnIII^ on efferocytosis mediated by Tim-4 and Mertk. Efferocytosis by cells expressing both Tim-4 and Mertk was superior to efferocytosis by cells expressing only Tim-4 or Mertk, as indicated by the percentage of cells that engulfed apoptotic cells. However, efferocytosis by cells expressing both Tim-4 and Mertk in the presence of GST-Mertk^FnIII^ was lower than that in the presence of GST (Figure 4D). Notably, the inhibitory effect of GST-Mertk^FnIII^ on efferocytosis by cells expressing both Tim-4 and Mertk was not due to decreased binding of apoptotic cells to phagocytes (Figure 4E), indicating that enhanced efferocytosis is due to the interaction between Tim-4 and Mertk, rather than alterations in the number of apoptotic cells binding to phagocytes. In addition, we tested the physiological relevance of the interaction between Tim-4 and Mertk on efferocytosis using peritoneal macrophages derived from *WT*, *Tim-4^-/-^*, and *Mertk^-/-^* mice. In agreement with a previous report, efferocytosis by *Tim-4^-/-^* or *Mertk^-/-^* peritoneal macrophages was inferior to efferocytosis by *WT* peritoneal macrophages. This inferior efferocytosis by *Tim-4^-/-^* or *Mertk^-/-^* peritoneal macrophages was not further impaired by GST-Mertk^FnIII^_,_ while efferocytosis by *WT* peritoneal macrophages was reduced and comparable with efferocytosis by *Tim-4^-/-^* or *Mertk^-/-^* peritoneal macrophages in the presence of GST-Mertk^FnIII^ (Figure 4F). Taken together, these data suggest that the interaction between Tim-4 and Mertk is indispensable for the enhancing effect of Mertk on Tim-4-mediated efferocytosis.

## 4. Discussion

The exposure of PS on the apoptotic cells and its recognition by receptor(s) on phagocytes has generated considerable interest. The identification of PS receptors such as Tim-4 and Bai1 advanced the comprehensive understanding of efferocytosis. Among PS receptors, Tim-4 has received the most attention. In the past decade, studies have revealed how Tim-4 recognizes PS on apoptotic cells and transduces signals into phagocytes during efferocytosis [18,19,21,22,26,33]. However, knowledge of Tim-4 signaling during efferocytosis is still incomplete. A recent finding by our group, namely, that a physical interaction occurs between the IgV domain of Tim-4 and the FnIII domain of Fn1, encouraged us to search for a transmembrane protein containing a FnIII domain that might interact with Tim-4 and transduce signals for Tim-4. Interestingly, Mertk comprises two Ig domains and two FnIII domains in its extracellular region. Moreover, it is known that Mertk functionally cooperates with Tim-4 to promote engulfment of apoptotic cells. However, a previous study was not able to detect a biochemical interaction between these two proteins [21]. This failure could be due to the use of antibodies recognizing the extracellular region of Tim-4 or Mertk; in this study, antibodies recognizing the extracellular region of Tim-4 or Mertk failed to co-precipitate Mertk or Tim-4, while antibodies recognizing the cytoplasmic tail of Tim-4 or a C-terminal tag of Mertk robustly co-precipitated Mertk or Tim-4, respectively.

As well as containing two FnIII domains, Mertk also contains two Ig domains, which are very similar to the IgV domain of Tim-4. Thus, the Ig domain of Mertk intra- or inter-molecularly interacts with its FnIII domain. This intra- or inter-molecular interaction of Mertk could be released by binding of either Tim-4 or bridging molecules such as Gas6 to Mertk. Furthermore, it is possible that Mertk binds to Fn1 through an interaction between the Ig domain of Mertk and the FnIII domain of Fn1, and joins to the previously identified Fn1–Tim-4–integrin complex to facilitate more efficient binding and ingestion of apoptotic cells. It will be interesting to investigate in future studies whether Mertk intra- or inter-molecularly interacts or forms a complex with the Tim-4–Fn1–integrin complex.

Taken together, results from our study suggest that the mechanism by which Mertk collaborates with Tim-4 during efferocytosis involves a biochemical interaction between the proteins. This interaction is essential for the enhancing effect of Mertk on Tim-4-mediated efferocytosis, and thus Mertk might directly aid Tim-4-mediated efferocytosis rather than participating in the so-called two-step engulfment process.

## Figures and Tables

**Figure 1 cells-09-01625-f001:**
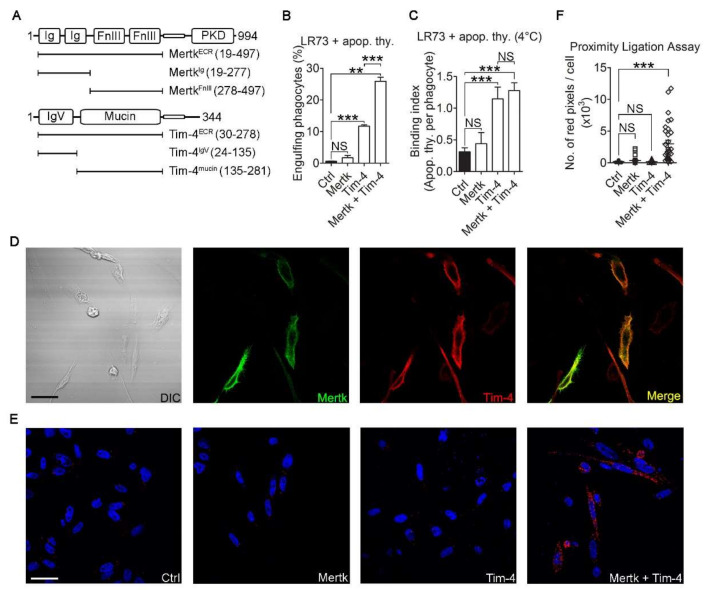
Tim-4 is colocalized with Mertk. (**A**) Schematic diagram of Mertk and Tim-4. Ig, immunoglobulin; FnIII, fibronectin type-III; PKD, protein tyrosine kinase domain; IgV, Immunoglobulin variable; ECR, extracellular region. (**B**) LR73 cells transfected with the indicated plasmids were incubated with TAMRA-labeled apoptotic thymocytes in serum-free alpha-MEM for 2 h, washed with PBS, trypsinized, and analyzed using flow cytometry. Cells that stained double-positive for TAMRA and GFP were considered to be phagocytes engulfing apoptotic cells (*n* = 4). (**C**) LR73 cells transfected with the indicated plasmids were incubated with TAMRA-labeled apoptotic thymocytes in serum-free alpha-MEM at 4 °C for 2 h and washed with PBS to remove unbound apoptotic cells. Bound apoptotic cells were then counted (*n* ≥ 50 cells). (**D**) LR73 cells transfected with HA-Tim-4 and Mertk-FLAG were incubated with anti-Mer and anti-HA antibodies, and then stained with Alexa Fluor 488- and Alexa Fluor 594-conjugated secondary antibodies. Images were acquired by confocal microscopy (*n* = 3). Scale bar, 20 µm. (**E**,**F**) A proximity ligation assay was performed using LR73 cells transfected with the indicated plasmids. One day after transfection, the cells were fixed, blocked, and incubated with anti-FLAG and anti-Tim-4^tail^ antibodies. The cells then were incubated with the amplification solution at 37 °C overnight. Images were acquired by confocal microscopy ((**E**), *n* = 3) and quantified ((**F**), *n* ≥ 27 cells). Scale bar, 40 µm. Data are shown as the mean ± standard deviation. NS, not significant. ***p* < 0.01. ****p* < 0.001.

**Figure 2 cells-09-01625-f002:**
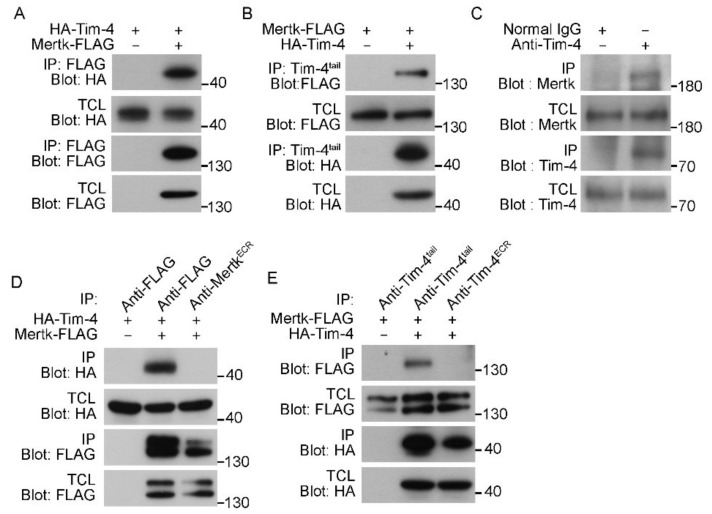
Tim-4 interacts with Mertk. (**A**) 293T cells were transfected with the indicated plasmids. Two days after transfection, the cells were lysed, and FLAG-tagged Mertk was precipitated with anti-FLAG antibody-conjugated agarose beads. Bound proteins were detected with the indicated antibodies (*n* = 7). (**B**) The lysates of 293T cells transfected with the indicated plasmids were incubated with an anti-Tim-4^tail^ antibody and protein A/G agarose beads. Bound proteins were detected by immunoblotting (*n* = 3). (**C**) The lysates of peritoneal macrophages were incubated with an anti-Tim-4^tail^ antibody or a control antibody and protein A/G agarose beads. Co-precipitated Mertk was detected by immunoblotting (*n* = 3). (**D**,**E**) 293T cells were transfected with the indicated plasmids and lysed. Mertk-FLAG ((**D**), *n* = 3) or HA-Tim-4 ((**E**), *n* = 3) in the lysates was precipitated with the indicated antibodies, and bound proteins were detected by immunoblotting. TCL, total cell lysate; IP, immunoprecipitation.

**Figure 3 cells-09-01625-f003:**
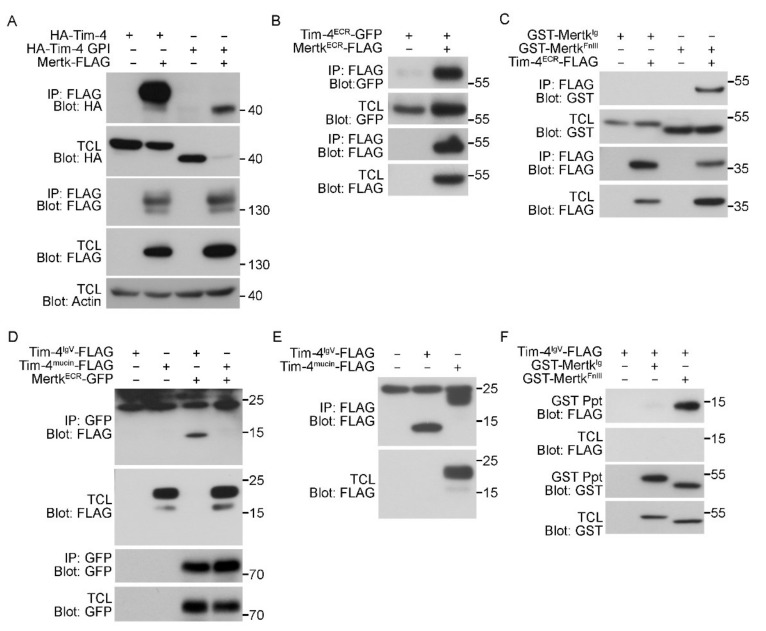
The IgV domain of Tim-4 binds to the fibronectin type III domain of Mertk. (**A**–**C**) 293T cells transfected with the indicated plasmids were lysed and incubated with anti-FLAG antibody-conjugated agarose beads. Bound proteins were detected by immunoblotting ((**A**), *n* = 3; (**B**), *n* = 3; (**C**), *n* = 3). (**D**) The lysates of 293T cells transfected with the indicated plasmids were incubated with an anti-GFP antibody and protein A/G agarose beads. Bound proteins were detected by immunoblotting (*n* = 3). (**E**) 293T cells transfected with the indicated plasmids were lysed and incubated with anti-FLAG-conjugated agarose beads. Bound proteins were detected by immunoblotting. Noticeably, the IgV domain of Tim-4 was only detected in immunoprecipitants but not in total cell lysates (*n* = 2). (**F**) 293T cells transfected with the indicated plasmids were incubated with glutathione-sepharose beads. Bound proteins were detected by immunoblotting (*n* = 3). TCL, total cell lysate; IP, immunoprecipitation.

**Figure 4 cells-09-01625-f004:**
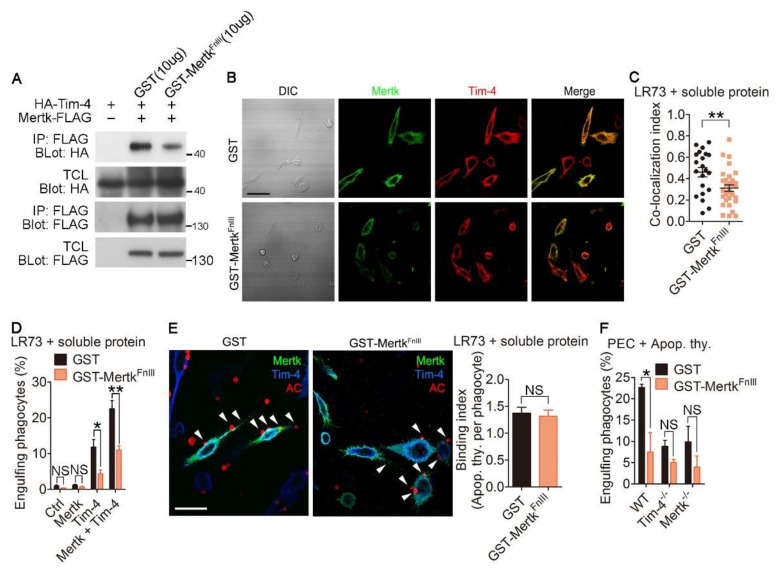
Disrupting interaction of Tim-4 with Mertk abolishes the synergistic effect of Mertk on Tim-4-mediated efferocytosis. (**A**) The lysates of 293T cells transfected with the indicated plasmids were incubated with anti-FLAG antibody-conjugated agarose beads in the presence of purified GST-Mertk^FnIII^ or GST. Bound proteins on the beads were detected by immunoblotting (*n* = 3). (**B**,**C**) LR73 cells were transfected with HA-Tim-4 and Mertk-FLAG. One day after transfection, the cells were incubated with purified GST-Mertk^FnIII^ or GST for 2 h, stained with anti-Mer and anti-HA antibodies, and labeled with Alexa Fluor 488- and Alexa Fluor 594-conjugated secondary antibodies. Images were acquired by confocal microscopy ((**B**), *n* = 2) and bound apoptotic cells were counted ((**C**), *n* ≥ 21 cells). Scale bar, 20 µm. (**D**) LR73 cells transfected with the indicated plasmids were incubated with TAMRA-labeled apoptotic thymocytes in serum-free alpha-MEM for 2 h in the presence of purified GST-Mertk^FnIII^ or GST, washed with PBS, trypsinized, and analyzed using flow cytometry (*n* = 4). (**E**) LR73 cells were transfected with HA-Tim-4 and Mertk-FLAG. One day after transfection, the cells were incubated with TAMRA-labeled apoptotic thymocytes in serum-free alpha-MEM at 4 °C for 2 h, washed, fixed, stained with anti-Mer and anti-HA antibodies, and labeled with Alexa Fluor 488- and Alexa Fluor 405-conjugated secondary antibodies (*n* ≥ 100 cells). Arrowheads indicate bound apoptotic cells. AC, apoptotic cells. Scale bar, 20 µm. (**F**) Peritoneal macrophages derived from *WT*, *Tim-4^-/-^*, or *Mertk^-/-^* were incubated with TAMRA-labeled apoptotic thymocytes in serum-free RPMI for 2 h in the presence of purified GST-Mertk^FnIII^ or GST, and then peritoneal macrophages engulfing apoptotic thymocytes were analyzed using flow cytometry (*n* = 4). Data are shown as the mean ± standard deviation. NS, not significant. ***p* < 0.01, **p* < 0.05. TCL, total cell lysate; IP, immunoprecipitation.

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
