# Peer review of "Mertk Interacts with Tim-4 to Enhance Tim-4-Mediated Efferocytosis"

_cells, 2020, doi:10.3390/cells9071625_

Round 1

Reviewer 1 Report

In this report, Moon et al. carry out a series of in vitro experiments to demonstrate that Tim-4, a well-known efferocytosis receptor (or co-receptor) mediates phagocytosis of apoptotic cells via interaction with another efferocytosis receptor, Mertk. Overall, the experiments are very well done, with appropriate controls and statistical analyses, and the manuscript is very well written and adds an important piece to the puzzle of how Tim-4 functions as an efferocytic receptor. Given the importance of efferocytosis in immunity and homeostasis, and the well-documented importance of Tim-4 in many disease settings, this work, although limited in scope, serves to advance the field in a meaningful way. I have only minor suggestions to make: 

Minor issues:

  1. Line 159: Authors should explain why serum-free conditions were used in these experiments.
  2. Fig1E: The authors should quantify the data from the proximal ligation experiments.
  3. General: the authors do not address the issue of Mertk phosphorylation and how that might impact Tim-4 binding. Was this looked at? Might be interesting to mention what is known about conformational changes in the Mertk ECR upon ligand binding (Gas6/ProS) and phosphorylation.
  4. Lines 194-196: This sentence was confusing. I think the authors are trying to say that (in Fig3A) HA-Tim4-GPI levels were always lower than HA-Tim-4 when Mertk was co-expressed.
  5. Lines 199-200: This sentence seems like a repeat of the previous sentence.
  6. Authors should add the n value for experiments in all the figures, especially for those where only representative data are shown (i.e. blots and photos).
  7. Fig4E: I’m not seeing many if any engulfment events in the GST control photo. Perhaps a better image can be selected and arrows used to point to engulfment events.

Author Response

We would like to thank the reviewers, who provided constructive reviews of the manuscript. Their specialized knowledge helped generate more reliable data, find more effective ways of expressing our conclusion, and thus improve the rigor and clarity of the manuscript. We are also grateful for the opportunity for revision.

Response to Referee #1

Reviewer 1

In this report, Moon et al. carry out a series of in vitro experiments to demonstrate that Tim-4, a well-known efferocytosis receptor (or co-receptor) mediates phagocytosis of apoptotic cells via interaction with another efferocytosis receptor, Mertk. Overall, the experiments are very well done, with appropriate controls and statistical analyses, and the manuscript is very well written and adds an important piece to the puzzle of how Tim-4 functions as an efferocytic receptor. Given the importance of efferocytosis in immunity and homeostasis, and the well-documented importance of Tim-4 in many disease settings, this work, although limited in scope, serves to advance the field in a meaningful way. I have only minor suggestions to make:

Minor issues:

Line 159: Authors should explain why serum-free conditions were used in these experiments.

In line 160-161 of the original manuscript, we briefly explained why we used serum-free conditions in the study. However, it seems that the explanation was not enough and we are sorry for that. Both Tim-4 and Mertk function as engulfment receptors promoting efferocytosis. However, the way of PS recognition by them is different. Tim-4 directly recognizes PS on apoptotic cells whereas Mertk indirectly recognize PS on apoptotic cells through bridging molecules such as Gas6 and Protein S. Thus, depending on the presence or absence of serum, the effect of Mertk on efferocytosis is different. Mertk is unable to mediate efferocytosis by itself in the absence of serum because it cannot bind to PS on apoptotic cells without a bridging molecule. Indeed, Mertk did not promote efferocytosis in the serum-free condition (Figure 1B in the original manuscript). In the study, we wanted to evaluate whether Tim-4 collaborates with Mertk during efferocytosis through a biochemical interaction with Mertk. To prove this, it is necessary to rule out the effect of Mertk by itself on efferocytosis. Thus, we performed efferocytosis assays without serum. We hope that the reviewer is satisfied with this explanation and the changes in the revised manuscript as follows. “We also confirmed the effect of Mertk on Tim-4-mediated efferocytosis under serum-free conditions, which did not contain bridging molecules such as Gas6 required for Mertk-mediated efferocytosis and thus excluded the effect of Mertk itself on efferocytosis.”

Fig1E: The authors should quantify the data from the proximal ligation experiments.

As the reviewer requested, we quantified the fluorescence intensity of cells in the proximal ligation assays. The fluorescence intensity in cells expressing both Tim-4 and Mertk was much higher than that in the other cells (Figure 1F in the revised manuscript).

General: the authors do not address the issue of Mertk phosphorylation and how that might impact Tim-4 binding. Was this looked at? Might be interesting to mention what is known about conformational changes in the Mertk ECR upon ligand binding (Gas6/ProS) and phosphorylation.

The kinase domain of Mertk is essential for efferocytosis because efferocytosis is impaired in MertkKD (Mertk kinase dead, PMID: 11346799) mice. It is also known that binding of its ligands or apoptotic cells through Gas6/ProS induces Mertk dimerization and phosphorylation (PMID: 18421305). These suggest that Mertk phosphorylation is required for Mertk-mediated efferocytosis. However, it is unclear whether ligand binding and phosphorylation induces a conformation change of the extracellular region (ECR). Nevertheless, as ligand binding induces Mertk dimerization by the ECR, a conformation change of the ECR is likely induced.

We found that Mertk overexpression induced phosphorylation as well. Thus, the IP experiments at overexpression conditions suggest that phosphorylated Mertk interacts with Tim-4. In addition, we also detected that Tim-4 and Mertk interact at endogenous protein levels which Mertk may not be phosphorylated at (Fig. 2C in the revised manuscript). To directly test whether Mertk phosphorylation could affect the interaction between Mertk and Tim-4, we treated cells with LDC1267, a Mertk inhibitor. In the presence of LDC1267, Mertk phosphorylation was drastically reduced, but the interaction between Mertk and Tim-4 was unaltered (Supplementary figure 1 in the revised manuscript). These data indicate that the interaction of Mertk with Tim-4 is independent on Mertk phosphorylation.      

Lines 194-196: This sentence was confusing. I think the authors are trying to say that (in Fig3A) HA-Tim4-GPI levels were always lower than HA-Tim-4 when Mertk was co-expressed.

We are sorry for the unclear sentence. The expression level of HA-Tim-4-GPI was decreased upon co-expression of Mertk-FLAG due to an unknown reason while the expression level of HA-Tim-4 was unaltered upon co-expression of Mertk-FLAG. Thus, we would like to say that the level of HA-Tim-4-GPI was decreased/lowered while the level of HA-Tim-4 was unaltered when HA-Tim-4-GPI or HA-Tim-4 was co-expressed with Mertk-FLAG. This is what we tried to say. To clarify the meaning of the sentence, as the reviewer suggested, we rewrote the sentence as follows. “HA-Tim4-GPI levels were always lower than HA-Tim-4 when Mertk was co-expressed.”  We thank the reviewer for clarifying the meaning of the sentence.

Lines 199-200: This sentence seems like a repeat of the previous sentence.

To avoid repetition, we revised the sentence as follows.

“Nevertheless, HA-Tim-4-GPI was robustly co-precipitated with Mertk by an anti-FLAG antibody (Figure 3A), suggesting that Tim-4 and Mertk interact through their extracellular regions”. Once again, we thank the reviewer for suggesting a more concise sentence.

Authors should add the n value for experiments in all the figures, especially for those where only representative data are shown (i.e. blots and photos).

As the reviewer requested,  we added the number of repeated experiments in all the figures.

Fig4E: I’m not seeing many if any engulfment events in the GST control photo. Perhaps a better image can be selected and arrows used to point to engulfment events.

Fig4E is a representative picture of the binding assay. Because phagocytes were incubated with apoptotic cells at 4 oC, apoptotic cells bound to phagocytes without ingestion. Thus, apoptotic cells shown in the picture are not engulfed but bound apoptotic cells. In addition, the experiment was performed at serum-free conditions, which resulted in a small number of apoptotic cells binding to phagocytes due to the absence of bridging molecules linking apoptotic cells to phagocytes. This is the reason why a relatively small number of apoptotic cells binding to phagocytes was observed in Figure 4E. We added arrowheads to indicate bound apoptotic cells. We hope that this explanation could relieve the reviewer’s concern.

Reviewer 2 Report

The manuscript "Mertk interacts with Tim-4 to enhance Tim-4-
3 mediated efferocytosis" deals with the biochemical interaction between the proteins Mertk and Tim-4 during efferocytosis. A manuscript fits into the scope of the journal and is of interest for the potential readers. The observations presented by the authors are very useful and interesting. The high-level methodology and quality of the study are worth stressing. To sum up, the manuscript presents interesting results and I recommend publication of this manuscript.

Author Response

We would like to thank the reviewers, who provided constructive reviews of the manuscript. Their specialized knowledge helped generate more reliable data, find more effective ways of expressing our conclusion, and thus improve the rigor and clarity of the manuscript. We are also grateful for the opportunity for revision.

Response to Referee #2

Reviewer 2

The manuscript "Mertk interacts with Tim-4 to enhance Tim-4-mediated efferocytosis" deals with the biochemical interaction between the proteins Mertk and Tim-4 during efferocytosis. A manuscript fits into the scope of the journal and is of interest for the potential readers. The observations presented by the authors are very useful and interesting. The high-level methodology and quality of the study are worth stressing. To sum up, the manuscript presents interesting results and I recommend publication of this manuscript.

We are pleased that the reviewer appreciates the true value of this study. We thank the reviewer for his/her nice comments on this study.

Reviewer 3 Report

The manuscript by Moon et al is a simple but interesting study with regard to two proteins putatively involved in efferocytosis. The experiments are largely done well and appropriate, and interpretation sound. However, given the kind of data used (immunoprecipitation and immunofluorescence) a couple of the figures could and should be improved to show how representative the data are of multiple experiments. After suitable revision, the manuscript should be acceptable for publication.

Concerns: Experiment 2C- The GST proteins are not expressed at uniform levels. In order to properly interpret the data the way the authors have in the manuscript, this experiment needs better equalization of protein between the samples.

Experiment 2 D,F. Because of the absence of the input bands in the IgV domain of Tim-4 (due to low expression), the authors need a better confirmation that the precipitated bands are the the correct bands. Use of another antibody giving uniform results would be helpful. This could either be a switched tag or a direct antibody to the antigen. Alternately, direct precipitation of the protein by FLAG antibody (such as in done in 4D), would be helpful if the proteins are run on the same gel

Experiment 4A- Since the reduction is modest, please include additional blots of the same experiment, to show that this is a reproducible effect.

Experiment 4B- The immunofluorescence pictures are not convincing and do not look significantly different. It is unclear as to how the quantitation was done in C (this information should be included, and more cells analyzed). I would ask that the authors  do this experiment using PLA (rather than regular immunofluorescence, as it is generally more quantitative for interacting proteins (especially of proteins in the same general localization area), and should be far more convincing that an actual reduction occurs.

Author Response

We would like to thank the reviewers, who provided constructive reviews of the manuscript. Their specialized knowledge helped generate more reliable data, find more effective ways of expressing our conclusion, and thus improve the rigor and clarity of the manuscript. We are also grateful for the opportunity for revision.

Response to Referee #3

Reviewer 3

The manuscript by Moon et al is a simple but interesting study with regard to two proteins putatively involved in efferocytosis. The experiments are largely done well and appropriate, and interpretation sound. However, given the kind of data used (immunoprecipitation and immunofluorescence) a couple of the figures could and should be improved to show how representative the data are of multiple experiments. After suitable revision, the manuscript should be acceptable for publication.

Concerns: Experiment 2C- The GST proteins are not expressed at uniform levels. In order to properly interpret the data the way the authors have in the manuscript, this experiment needs better equalization of protein between the samples.

We think that the indicated figure by the reviewer is not Figure 2C but Figure 3C. In order to uniformly express the GST tagged proteins, we adjusted the amount of the plasmids for transfection and performed the IP experiment again, and the levels of the GST-fusion proteins were comparable although there was a slight difference of expression level between GST-MertkIg and GST-MertkFnIII. The extracellular region of Tim-4 still showed robust interaction with the FnIII domain but not the Ig domain of Mertk. We replaced the old blots with the new blots (Figure 3C in the revised manuscript)

Experiment 2 D,F. Because of the absence of the input bands in the IgV domain of Tim-4 (due to low expression), the authors need a better confirmation that the precipitated bands are the the correct bands. Use of another antibody giving uniform results would be helpful. This could either be a switched tag or a direct antibody to the antigen. Alternately, direct precipitation of the protein by FLAG antibody (such as in done in 4D), would be helpful if the proteins are run on the same gel.

Once again, we think that the indicated figures by the reviewer is Figure 3D and F. First, the size of the precipitated band in the IP blot exactly coincided with the size of Tim-4IgV-FLAG, which was also confirmed in our previous study (PMID: 30451988). Second, as the reviewer requested, we performed experiments with the sample that is direct precipitation of Tim-4IgV-FLAG. Again, the co-precipitated band with MertkECR-GFP coincided with the band of precipitated Tim-4IgV-FLAG with an anti-FLAG antibody (Supplementary figure 2 in the revised manuscript). We thank the reviewer for suggesting a better way of presentaiton.

Experiment 4A- Since the reduction is modest, please include additional blots of the same experiment, to show that this is a reproducible effect.

The inhibitory effect of soluble GST-MertkFnIII on the interaction between Tim-4 and Mertk was reproducible and substantial although it did not completely abrogate the interaction. As the reviewer requested, we added another blot showing that GST-MertkFnIII inhibits the interaction between Tim-4 and Mertk (Supplementary figure 3 in the revised manuscript). In addition, we showed the inhibitory effect of soluble GST-MertkFnIII on the interaction using three different approaches (IP, co-localization, and PLA (a new approach used in revision)). Results from the approaches suggest the notable inhibitory effect of soluble MertkFnIII on the interaction of Tim-4 with Mertk. 

Experiment 4B- The immunofluorescence pictures are not convincing and do not look significantly different. It is unclear as to how the quantitation was done in C (this information should be included, and more cells analyzed). I would ask that the authors  do this experiment using PLA (rather than regular immunofluorescence, as it is generally more quantitative for interacting proteins (especially of proteins in the same general localization area), and should be far more convincing that an actual reduction occurs.

We calculated the co-localization index as follows. The co-localization index formula is the number of pixels with a colocalization color divided by the number of total pixels. The range of two colors from green (Mertk) to red (Tim-4) was set, and the pixels in the middle range of the color (yellowish) was counted using ImageJ (Supplementary figure 4 in the revised manuscript). This method is generally used to calculate the co-localization index. We added this method in the materials and methods part of the revised manuscript. In addition, as the reviewer requested, we performed the PLA assay in the presence of GST or GST-MertkFnIII.  The PLA fluorescence from cells expression Tim-4 and Mertk in the presence of GST-MertkFnIII was noticeably lower than that from control cells (Supplementary figure 5 in the revised manuscript), suggesting that GST-MertkFnIII impedes the interaction between the proteins. Once again, the effect of soluble GST-MertkFnIII on the interaction between Tim-4 and Mertk was evaluated using various approaches. We hope that various approaches relieve the reviewer’s concern.
